# How Do Young Adult Drinkers React to Varied Alcohol Warning Formats and Contents? An Exploratory Study in France

**DOI:** 10.3390/ijerph20156541

**Published:** 2023-08-07

**Authors:** Gloria Thomasia Dossou, Morgane Guillou-Landreat, Loic Lemain, Sophie Lacoste-Badie, Nathan Critchlow, Karine Gallopel-Morvan

**Affiliations:** 1LUMEN (ULR 4999), ILIS, Faculty of Engineering and Health Management, University of Lille, 42 rue Ambroise Paré, 59120 Lille, France; 2EA 7479 SPURBO, School of Medicine, University Bretagne Occidentale, 5 Avenue Camille Desmoulins, 29200 Brest, France; morgane.guillou@chu-brest.fr (M.G.-L.); loic.lemain@chu-brest.fr (L.L.); 3LUMEN (ULR 4999), IAE Lille, University School of Management, University of Lille, 104 Av. du Peuple-Belge, 59000 Lille, France; sophie.lacoste-badie@univ-lille.fr; 4Institute for Social Marketing and Health, University of Stirling, Stirling FK9 4LA, Scotland, UK; nathan.critchlow@stir.ac.uk; 5CNRS, Inserm, Arènes-UMR 6051-U 1309, EHESP, School of Public Health, University of Rennes, 15 Av. du Professeur Léon Bernard, 35043 Rennes, France; karine.gallopel-morvan@ehesp.fr

**Keywords:** alcohol, warnings, France, drinking culture, young adults

## Abstract

Research on alcohol warnings has increased in the last decade, providing key evidence to governments on warning format and contents. The bulk of this research, however, has been conducted in Anglosphere countries, whereas fewer studies have focused on other countries which have high per capita alcohol consumption, and where the high social acceptability of drinking is liable to affect how people accept and react to prevention measures. Since France has one of the highest per capita alcohol consumption rates in the world according to the World Health Organization (WHO), we therefore explore how young adults in France react to warnings on alcoholic beverage advertisements. We conducted 25 in-depth interviews, in 2017, with 18–25-year-old drinkers in France. Respondents were asked open-ended questions on the perceived impact of various warning contents (i.e., on health risk, social-cost risk, and on short- vs. long-term risk) and formats (text only vs. larger text combined with colored pictograms). Warnings that targeted youth-relevant risks (i.e., road accidents or sexual assault) were considered to be the most meaningful and credible, although warnings communicating longer term risks (i.e., brain, cancer) were also thought to be influential. Less familiar risks, such as marketing manipulation and calorie intake, elicited the most negative reactions. Larger text-and-pictogram warnings were considered to be the most effective format in capturing attention and increasing awareness. Regardless of format and content, however, these warnings were not perceived as effective for decreasing alcohol consumption.

## 1. Introduction

Alcohol use causes about three million deaths per year worldwide [1]. Young adults are one of the most vulnerable populations, since alcohol use in young adults is associated with violence, accidents, suicide, and a myriad of other consequences [2]. France, where this research takes place, ranks among the top 10 countries in Europe for alcohol consumption per capita (11.4 L per year per capita, in 2019, for those aged 15 and over [3]) and registers 41,000 alcohol-related deaths per year [4]. Youth drinking is a major public health issue; about 50% of French people aged 18–34 years old report at least one heavy drinking episode per month [5]. While some countries have implemented a robust range of alcohol control policies in the last decade (Lithuania, Russia) [6,7], France lags behind; France has no national program on alcohol control and no evidence-based measures on alcohol taxation, and its current laws on marketing restrictions (Évin law [8]) and the ban on sales to minors (under 18 years old) are not effectively enforced [9,10]. In the meantime, intensive lobbying by the French alcohol industry [11] manages to spread alcohol-positive messages via mainstream media and opinion leaders, including French President Emmanuel Macron who says he drinks wine “daily, lunch and dinner” and sees wine as “inseparable from our art of living, the art of being French” [12,13]. The upshot of this combination of weak alcohol control policies and strong industry influence is that France still has a high per capita alcohol consumption and high social acceptability of drinking alcohol [14,15]. Previous research has shown how sociocultural context is an important determinant of attitudes to drinking and alcohol-related policy [16,17,18]. For instance, Bocquier et al. [19] and Annunziata et al. [15] found that French people tended to downplay, deny, and/or underestimate the risks associated with alcohol consumption. Regarding alcohol warnings displayed on containers and/or in ads, which is a policy measure recommended by the WHO [20], Andrews et al. [21,22] highlighted that people who had a favorable attitude towards drinking and/or people who drink frequently were less likely to believe labeling messages. To expand ongoing studies, our research explores how young adult drinkers in France react to various alcohol warning formats and contents. France currently imposes two mandatory health warnings on alcohol, i.e., a statement that reads “Alcohol abuse is dangerous for your health” displayed at the bottom of all advertisements for alcoholic beverages since 1991, and a small pictogram representing a pregnant woman drinking alcohol that is displayed on all alcoholic beverage containers since 2007. These two warnings, however, have been considered to be ineffective at capturing attention and informing young adults in France, as they are not prominent enough, too small, too old, and the text is unclear [23]. Therefore, there is a need to change these messages to more effective ones, and therefore, to address this need, we set out to explore and better understand how young adult drinkers living in a country such as France, where drinking alcohol is socially acceptable, react to various new warning contents and formats, and which warnings are most effective to target them.

The second aim of this research is to contribute to the literature on alcohol health warnings more generally. First, most of the extant research on this topic has been performed in Anglosphere countries (e.g., the United States, United Kingdom, Australia, and Canada). So far, there has been limited researched in EU countries, whereas the European Commission has proposed to bring in mandatory health warnings on alcoholic beverages by the end of 2023, as part of the Beating Cancer Plan [24] (p. 10). Second, recent evidence reviews have provided recommendations for increasing the effectiveness of alcohol warnings in general [25,26,27], but there is ongoing debate surrounding which warning contents and formats are most effective in targeting young people. Some research has concluded that exposure to alcohol warnings reduces the appeal and social acceptability of alcohol products [28] and significantly increases knowledge about alcohol-related risks among the youth segment [29], whereas other studies have found that alcohol warnings tended to elicit negative and ”on-the-defensive” reactions in young people that could prove to be counterproductive or even increase binge-drinking behaviors [21,30]. Some studies have shown that messages featuring short-term (rather than long-term) alcohol risks raise greater awareness of the harms of alcohol among young people [31] because young people feel particularly concerned by short-term risk warnings [32,33,34]. Other studies, however, have found that long-term risk warnings about cancer can also be youth-relevant messages as they generate fear [35,36]. There is also evidence that messages focused on social consequences of alcohol use that affect friends (i.e., traffic accidents) and on cognitive consequences (ability to concentrate) are more effective than messages concerning health risks (coma and death) among young drinkers [37]. Hassan et al. [38], however, found that warnings on social consequences of alcohol use were not particularly persuasive among students. Beyond warning content, Vallance et al. [39] looked at warning formats and showed that combining a pictogram with a text message increased attentional capture and facilitated comprehension among young people, and Jones et al. [40] studied a sample of young adults in the UK and found that warnings in novel forms (text only, text and image) were the most engaging and potentially effective. In line with these results, Annunziata et al. [32] found that adding pictograms positively affected reactions but did not influence intended alcohol-related behavior among a young adult target audience. In 2022, young Mexican adults were presented with the image of a conventional beer featuring pictograms (red font on a white background, or with a black font on a yellow background), located at the top and wrapping around a third of the beer can. The study found that these visible health warning labels could lead individuals to reflect on the health risks of alcohol, reducing the appeal of the product and lowering the intention to buy and consume alcohol [41].

In short, research into alcohol warnings that could effectively target young adults has led to globally divergent conclusions and inconsistent results. To address this, our study aims to explore and better understand how young adults living in a drinking-conducive culture (France) react to various new-content and new-format alcohol warnings. We designed this study to address the following research questions:How do young adults, who are living in a context where social acceptability of alcohol consumption is high, react to various warning contents and formats?Are some warning contents (social risks, short-term risks, etc.) more relevant than others for speaking to young adults?Is there added benefit in combining a text message with a pictogram?

## 2. Materials and Methods

We employed a qualitative research design based on individual face-to-face interviews. This strategy enabled participants to freely express their opinions on a given subject without the peer-pressure influences which could emerge in focus groups, especially in France which continues to enjoy “positive” cultural and social norms around alcohol consumption. 

### 2.1. Recruitment, Sample, and Instruments 

Twenty-five drinkers, aged 18–25 (13 women), who lived in Brittany (France) were recruited by a market agency (IRS) from an existing panel. They were invited via email to share their views and opinions on alcohol consumption (they were informed that the research was not being conducted by the alcohol industry), but the topic of warnings was not presented at this stage. Respondents were asked to give their age, gender, occupation, and alcohol consumption as part of the recruitment process. The Alcohol Use Disorders Identification Test-Concise (AUDIT-C) was used to define their drinking status as “non-drinker”, “moderate drinker” or “high-risk drinker” [42]. 

The AUDIT-C is a three-point scale that measures the following: (1) frequency of drinking alcoholic beverages (scored from 0 = never to 4 = four or more times per week), (2) number of alcohol units consumed in a typical occasion (scored from 0 = one or two units to 4 = ten or more units), (3) frequency of consuming six or more alcohol units on a single occasion (scored from 0 = never to 4 = every day or so) [42]. The scores for each response were tallied to give an overall score that corresponded to a drinking status (moderate risk for alcohol use disorder = <2 for women and <3 for men; high risk for alcohol use disorder = between 3 and 6 for women and between 4 and 7 for men). Non-drinkers were not surveyed. People who scored higher than 7 (women) and 8 (men) were excluded from the survey as they were likely at risk for alcohol use disorder. The subsequent sample, therefore, comprised 13 moderate and 12 high-risk drinkers. 

In summary, the inclusion criteria were age (between 18 and 25), drinking status (moderate and high-risk drinkers according to the AUDIT-C test), language skills (fluency in French, to clearly understand the messages), and sector of activity (not working for the alcohol industry or studying in this field, to avoid conflicts of interest). At the end of the interviews, respondents signed a form giving consent to use and analyze their responses, and received a 20 Euro gift voucher as compensation for their participation. Table 1 describes the profile of the 25 participants and their defined drinking status.

### 2.2. Study Materials

We created 12 novel alcohol warnings for France (Figure 1): nine messages featured classic health and social-cost risks (e.g., increased risk for cancer, alcohol coma, alcoholism, or car accidents), and three featured ”innovative” prevention messages that they are rarely used in public health campaigns (risks of excess calorie intake, environmental impact, manipulation by the alcohol industry). All selected risks were evidence based, with information and statistics found via official websites (French Government, public institutes, non-governmental organizations) or in previous research. The three ”innovative’ messages were selected on the rationale that their contents were youth relevant who have rarely been investigated [27]. The calorie intake message (“2 glasses of beer = 1 burger = 250 calories”) was selected because Bollinger et al. [43] and Campos et al. [44] suggested, in a non-alcoholic beverage context (sodas), that calorie intake labeling might decrease consumption. The message that read “Wine may contain carcinogenic pesticides” was chosen because young people tend to be environmentally conscious and have ”green’ purchasing behaviors [45,46]. The message “The alcohol industry manipulates you thought advertising. Don’t be fooled!” was tested because such counter-marketing messages have been widely used in the context of tobacco (e.g., the “truth” campaign [47]), and have been shown to be effective in decreasing smoking among young people. The persuasiveness process behind this message is explained by the inoculation theory, which posits that people can be protected against attempts at commercial manipulation if they are warned against them with counter arguments [48]. Each of the 12 text-format messages was paired with a color pictogram portraying the risk. The pictograms were found on the Internet and/or redesigned by a professional (as necessary) to fit the text-message content.

Figure 1 presents the 12 tested text message and pictogram warnings. These were classified according to previous research [37,49] as being either short-term or long-term alcohol risks and as either health-cost or social-cost warnings. The 12 warnings were presented to respondents in different formats. First, participants were asked to compare three formats: the current (small) French format vs. the larger text-only warning format vs. the larger text-plus-pictogram format (Figure 2A). Second, participants were asked to focus on the larger text-only warning format and the larger text-plus-pictogram format (Figure 2B) and to give their perceptions by comparing these two formats. All warnings were presented to respondents on printed boards.

**Figure 1 ijerph-20-06541-f001:**
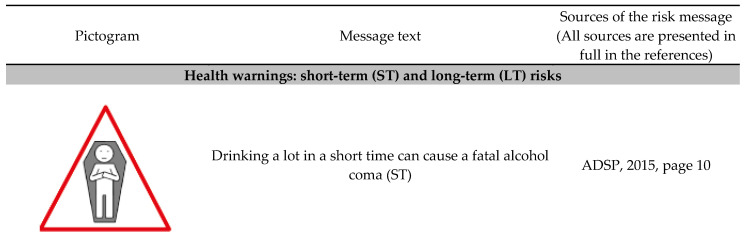
The 12 warning messages and associated pictograms. **Legend:** All sources are presented in full in the following references: risk of coma [50]; risk of cancer [51,52]; risk of brain damage [37,53]; risk of alcoholism [54]; risk of disability in babies [37]; risk of drunk-driving [55]; risk of violence [56]; risk of sexual assault [56]; risk of isolation [57]; risk of calories intake [56]; risk of manipulation [58]; risk of carcinogenic pesticides [56].

### 2.3. Ethics Approval

This research received ethical approval from the French National Cancer Institute (ASAFE Project No. 2017-109 “INCa_11913” Institut National du Cancer”-National Cancer Institute), and this study conforms to the Declaration of Helsinki principles and CNIL (the French data privacy agency) guidelines.

### 2.4. Procedure and Analysis

The interviews took place in a French university during June 2017. Participants were welcomed to the university, put at ease, and offered non-alcoholic beverages. Then, they were reminded of the purpose of the study (“discussion around alcohol consumption”) and told that the interviews would be recorded. 

The interviews lasted an average of one hour and were divided into four parts: (1) After presenting themselves, participants were asked questions on alcohol in general (e.g., perceived risks, benefits, consumption) and on the current alcohol warnings used in France (e.g., have they noticed them, what was their opinion); results on this first part are not presented here due to place and concision constraints). (2) Participants were randomly (to avoid a halo effect) presented the 12 new-content text-only warnings (Figure 1). (3) Participants were shown the warning on drink-driving risks displayed on real advertisements for alcoholic beers in different formats, i.e., the current French format, a larger text-only warning, a larger text-plus-pictogram format (Figure 2A). (4) Participants were exposed to the 12 new warning contents displayed on real advertisements presented in two formats, i.e., large text-only format and large text-plus-pictogram format (Figure 2B).

To understand the whole process and general reactions to alcohol messages and to predict their effective persuasiveness, participants were interviewed using the framework developed by Noar et al. [59] to analyze the effect of tobacco warnings: attention to warning messages, cognitive reactions (beliefs, susceptibility), affective reactions (positive or negative emotions), credibility, intention to cut down or stop drinking. They were also asked to give their opinion on which message warnings they thought were the most effective. At the end of the interview, we re-collected data on age, gender, occupation, education, and alcohol consumption. 

Data saturation was reached by the completion of the 25th participant (i.e., no new reactions emerged regarding the initial thematic framework), and so we stopped recruitment and data collection at this point [60].

Each recorded interview was transcribed in full, and a thematic content analysis was carried out manually by one researcher using the NVIVO 12 software (All figures/maps from NVIVO can be made available on request). Thematic analysis is a method that involves interpreting the content of a text by identifying a pattern of concepts [61]. Here, the thematic analysis was deductive [62] as it was based on the variables identified in the framework developed by Noar et al. [59].

## 3. Results (The Main Results by Warning Topic Are Summarized in Table 2)

### 3.1. Overall Reactions to the 12 (Text-Only) New Warning Contents

Most respondents welcomed the idea of replacing the current French warning displayed on ads (“alcohol abuse is dangerous for health”) with more risk-specific messages, as they found that most of new messages were strong, impactful, and appropriately youth targeted: 


*“… I think it’s a good idea to talk about alcohol coma, and adding the word ‘fatal’ drives the message home even more. It’s true, you can die from that, so I think it’s good idea…”*
(MMD, 19 (MMD, 19: Male moderate drinker, 19 years old))

They spontaneously suggested these messages should be rotated:


*“The idea of rotating the messages is important; that way, you get to target everyone”*
(MHRD, 25 (MHRD, 25: Male high-risk drinker, 25 years old))

The majority of participants understood the main idea behind most of the warnings. However, they stated that wording such as “*drinking a lot*”, “*excessive consumption*”, “*short time*”, or “*too much*”, which were used in some messages, was too imprecise and should be made more specific:


*“I think we don’t realize what it is: drinking a lot, drinking regularly, or drinking a lot in one session.”*
(FMD, 18 (FMD, 18: Female moderate drinker, 18 years old))

To enhance effectiveness and trigger negative effects, they also suggested details of the consequences of alcohol use on each organ, giving more information on quantity and frequency of drinking, and adding figures (for example, drunk driving message).

Most participants conceded that the 12 new message contents increased their awareness of certain specific alcohol-related risks (e.g., cancer, alcohol coma, brain damage):

[risk of cancer] *“No, I did not know that. Well, at least in low doses; I know that for an alcoholic, there are necessarily risks, but in low doses…”*(FHRD, 25 (FHRD, 25: female high-risk drinker, 25 years old))

However, they were skeptical about the impacts the message would have on drinking, for the following reasons: (i) uncertainty that the risk will arise, (ii) lack of novelty of some new messages, (iii) a general overemphasis on prevention (i.e., against tobacco, against unhealthy food), and (iv) a warning alone being not enough to trigger real behavioral change:


*“It’s hard to say whether one sentence alone will make you stop drinking.”*
(FMD, 19)

Some critical reactions to the new warning contents emerged, mostly underestimation of the risks tied to alcohol consumption and the threat to freedom:

[risk of alcoholism] *“When you’re out with friends, you might drink ‘before’—at the bar or at home— and then in a nightclub, and then during the ‘after’—back at home. But does that make you an alcoholic? No, I don’t think so.”*(MHRD, 24)

### 3.2. Reactions to Health Warnings and Social Warnings 

Health warnings appeared to improve young people’s knowledge and awareness compared to social-cost warnings:

[risk of coma] *“Yeah, well yeah, sure it concerns me, uh, you don’t want to try it when you see that, when you see the word ‘fatal’ in any case, it doesn’t make you want to play around with that”*(MMD, 24)

[Brain risk] *“I find that it’s like there’s a taboo around saying it damages the brain… it’s true that the effects on the brain, you don’t see that used much in alcohol awareness campaigns or when we talk about the risks we might face.”*(MMD, 19)

Regarding content credibility, as participants had already experienced most of the social risks presented, they were less dubious about their believability:

[drunk driving risk] *“…it affects me personally, because I lost a friend not long ago because of that.”*(MHRD, 20)

[sexual risk] *“it could happen to any of us.”*(FHRD, 19)

Participants highlighted that the risks presented in some messages were little known (e.g., fatal alcohol coma, risk of cancer, and brain damage), and this made them question the credibility:


*“I’ve never heard that alcohol coma can be fatal. I think that’s fake news.”*
(FHRD, 19)

Health-risk warnings and social-cost warnings were both perceived as personally relevant to informants, as most of the message contents were judged to be risks that young adults are specifically exposed to:


*“It brings back memories; I have a friend who went into an alcohol coma. It’s very real, and I think there are too many people that might not know that, or are not overly concerned … but it can happen really quickly”*
(FHRD, 19)

Even though they feel concerned, participants considered that these warnings would more likely be effective for targeting young teenagers who start drinking:


*“It can still speak more to young people… I think that after a certain time, you get more careful, you know your limits. For young people who are starting drinking (I would say 14 to 20 years old) and who start drinking way too much sometimes, it might make those specific people think about it a little more.”*
(FMD, 25)

Concerning behavioral effects, although health-risk and social-risk messages prompted young people to reflect on their drinking patterns, they were perceived as not being enough to motivate them to drink less in general, except for social-risk message contents that performed better on cautionary behavior (i.e., having to be escorted to avoid sexual assault due to drunkenness) and for the drunk driving context:


*“It’s extra information, it’ll help me talk about it and avoid drunk driving.”*
(FMD, 22)

[sexual risk] *“…there’s a debate to be had, for sure… like reducing the risk, or at least maybe more often trying to find a ride home, or have a friend drive you home. Or take a cab, or whatever. Maybe think about that already.”*(MHRD, 24)

### 3.3. Reactions to Long-Term and Short-Term Risk Warnings

Whatever the risks (health or social cost), warnings highlighting long-term issues were generally perceived as less relevant than warnings highlighting short-term risks for both the respondents themselves and other young people:


*“Young people don’t think about alcoholism at all, they think that only people older than 40 or 50 can become alcoholics, so I think they just don’t realize.”*
(MHRD, 20)

The persuasiveness of long-term risk messages, however, seemed to depend on their content and temporality. For instance, the young respondents perceived fetal alcohol syndrome as a relevant message, maybe because the risk is not too far down the road for some of them:


*“It will mark all pregnant people and make people aware that you can destroy the life of another person, your child.”*
(MHRD, 20)

Participants felt different threats around some long-term risks:

[brain risk] *“And focusing on the brain is good, because we care about it… So the message will tell everyone about the brain risk, and that’s going to concern more people than alcohol coma.”*(MMD, 22)

[risk of cancer] *“I say, everything leads to cancer today, like genes… Anyway, we all have cancer-causing genes in our bodies, so basically it all depends on whether you get lucky and those genes are not triggered.”*(FHRD, 19)

Participants perceived other long-term risks as being less effective messages, either because the risks did not directly relate to the participants’ current circumstances, or because they were considered to be too far into a hypothetical future:

[risk of isolation] *“alcohol connects young people closer to their peers. So, this warning will only be relevant to old people”*(MMD, 22)

[risk of alcoholism] *“…I don’t think it’s really going to speak to young people either, of if it does, it’ll be more in the long term…”*(MHRD, 19)

Short-term risks generated more homogeneous reactions in terms of their perceived effectiveness because participants had already experienced these risks (road accident, sexual assault, alcohol coma) and so they felt more directly concerned:


*“In our generation, everyone knows someone who was killed in a road accident. It means something to us.”*
(FMD, 25)

[risk of coma] *“In high school it was all about drinking for the sake of drinking and to get drunk… in high school, it’s about drinking a lot in a short burst, I think that’s what a lot of high-schoolers do.”*(FHRD, 19)

### 3.4. Perceptions of the Three “Innovative” Warnings 

Some participants, albeit not the majority, found the messages about calorie intake, environmental issues, and manipulation by the alcohol industry easy to understand and helpful in increasing knowledge and awareness of these issues:

[calorie intake] *“… I always thought that alcohol didn’t make you fat, I don’t know why […]. When I was dieting, drinking alcohol didn’t stop me.”*(MHRD, 20)

[marketing manipulation] *“…[this warning] would encourage people to take a more critical look at the advertisements…and be more skeptical about advertising in general… I think that young people, more than other groups, don’t like to feel they are being manipulated.”*(FMD, 18)

However, many respondents did not welcome these warnings. They generated numerous negative reactions such as being perceived as ridiculous, irrelevant, exaggerated, preachy, odd, unbelievable, unengaging, or not specific enough:

[calorie intake] *“Me personally, it’s going to make me laugh more than anything else: we usually drink a beer with a burger anyway, so we’re blowing the calories.”*(FHRD, 20)

[pesticides]: *“There are pesticides everywhere, everything is carcinogenic. You might never smoke in your life and still get lung cancer… My mom is in there…”*(FHRD, 19)

Some warnings and risks were also perceived as being easy for people to avoid and prevent without having to change the way they drink:

[pesticides] *“This might lead to people maybe drinking more of the organic wines that are certified pesticide free.”*(MMD, 22)

[calorie intake] *“I know beer makes you fat, you just have to work out to burn off the calories, that’s all…”*(MHRD, 24)

The message on manipulation by the alcohol industry generated a lot of criticism and even prompted a ”boomerang” effect:

“*I don’t really get influenced by advertising. So, I mean, I don’t find that advertising by manufacturers manipulates us, because drinking is already part of our culture…”*(MHRD, 19)

### 3.5. The Benefit Value of a Larger Warning Format COMBINED with Pictograms

Almost all participants found that the current warning format used in France failed to effectively capture attention and had little overall effect. Over the course of the interviews, and before presenting the pictogram format, some participants spontaneously suggested that adding images to a text message would increase its impact:


*“For me, a slogan like that, without … without some kind of image or video to support it, it doesn’t have a huge impact…”*
(MMD, 22)

When respondents were asked to give their opinion on large text-only warning format vs. the large text-plus-pictogram format, they almost universally pointed to the larger font size, the black-on-white warning, the use of a pictogram, and the novelty of the message content as making it more readable and more likely to quickly capture their attention:


*“You can’t ignore it… I think the idea of the big white banner, which is quite wide, with the pictograms would be very effective if it was put in place.”*
(FMD, 19)


*“We know we would pay more attention to it because, with the pictograms, we’re more vigilant about drugs, advertising, traffic laws, the pictograms speak more.”*
(MMD, 25)

Participants also cited comprehension and literacy as reasons explaining the benefit of adding pictograms to alcohol warnings:


*“[…] the pictogram with the poster is easier to see… you can understand it without necessarily reading the sentence.”*
(FHRD, 20)


*“… It can make people react, even children who see it might think… When you’re a child […] just by seeing the image, children who can’t necessarily read or anything, it can be good.”*
(FHRD, 25)

The pictorial format was also perceived as relevant for targeting young adults; it increased awareness of alcohol-use risks (by lending greater realism to the risk depicted), and was more engaging for some respondents:


*“Here it shows you the thing, you know. Someone hitting somebody else; you have heard about it already, but here you’re forced to see it.”*
(MHRD, 24)

Alongside these positive effects, the pictograms elicited few negative reactions, except certain pictograms (depicting risks of cancer, brain damage, alcohol coma, and pesticides) that were considered to be unclear by some respondents:

Concerning behavioral intentions, only a few participants felt that the pictograms could trigger behavioral change (motivate people to reduce their alcohol consumption, prompt pregnant women to abstain, or discourage underage drinking) by generating emotional reactions among specific targets and creating awareness:

[pregnancy pictogram] *“I think it’s really good, because maybe there are a lot of women who think ‘if I drink, it won’t reach the baby, it won’t get through’, but of course it will.”*(FHRD, 19)

[pictogram of sexual risk] *“It could scare kids away from drinking…”*(FMD, 22)

Most respondents felt that pictograms would have little or no effect on behaviors, because compared with photos, they are not realistic and shocking enough:


*“For me, [the pictogram] is clearer, but a real picture would work better”*
(FMD, 21)


*“…For me, it’s really photos that turn you off… they can disgust you too”*
(MMD, 22)

Among all the pictograms, the image depicting car accidents was perceived as the most effective for changing behaviors, due to various content-related and format-related reasons: easy to understand, feeling directly concerned (already confronted with the risk), youth relevant (citing the age of 18–24 target audience), and use of statistics (‘*1 out of 4′*):


*“It’s good, it has all the ingredients: the car, the broken bottle, and the person targeted by the message.”*
(MMD, 22)

**Table 2 ijerph-20-06541-t002:** Main results.

Warnings	Participants’ Reactions			
	Attention *	Cognitive Reactions	Credibility	Intentional Behaviour
Content				
Health warnings Coma risk Cancer risk Brain damage risk Alcoholism risk Fetal alcohol syndrome	N/A	Some health warnings improve participants’ knowledge and awareness (e.g., cancer risk, brain damage). Some health warnings were perceived as personally relevant to informants (e.g., coma risk, brain damage).	Some of the risks presented are little known (e.g., fatal alcohol coma, risk of cancer, and brain damage), which made some participants question their credibility.	Perceived as not being enough to motivate participants to drink less. Prompted young people to reflect on their drinking patterns.
Social-cost warnings Drinking and driving risk Violence risk Sexual assault risk Isolation risk	N/A	Participants felt concerned because they are specifically exposed to these risks. Most social risks were perceived as personally relevant to informants.	Participants were less dubious about the credibility of most social-cost warnings except isolation risk	Prompted young people to reflect on their drinking patterns. Perceived as more likely to be effective for targeting young teenagers who start drinking. Trigger cautionary behavior (e.g., having to be escorted to avoid drunk driving or sexual assault due to drunkenness).
Innovative warnings Calorie intake Marketing manipulation Pesticides	N/A	Some participants (but not the majority) found the messages easy to understand and helpful in increasing knowledge and awareness of these issues. Most respondents found these warnings ridiculous, irrelevant, exaggerated, etc.	Warnings perceived as unbelievable.	Perceived as easy to avoid and prevent without having to change how people drink.
FORMAT				
Benefit Value of a Larger Warning Format Combined with Pictograms	Quickly capture people attention. Make the warning more readable.	Improve warnings understanding and reduce literacy issues. Increase alcohol-use risks awareness. Give a greater realism to the risk depicted. Make the warning more engaging for some respondents.	Compared with photos, pictograms are not considered to be realistic and shocking enough.	Pictograms would have little or no effect on behaviors.

***** Attention has been assessed for warning format only.

## 4. Discussion

The aim of this research was to explore young adult drinkers’ (18–25 years) reactions to various and novel alcohol warning contents and formats in a European country (France) where the social acceptability of drinking is high. Our results show that most young people interviewed welcomed the idea of replacing the current French warning displayed on advertisements with new warnings that convey more youth-relevant messages, especially messages on specific risks to which young people are exposed. Some participants suggested a need to rotate the different warning labels for greater effect. 

We identified several factors that influenced the potential effectiveness of alcohol-use warnings for young adults, including: (i) the warnings refer to short-term rather than long-term risks, (ii) whether the audience feels susceptible to the risk, (iii) whether the risk depicted is already known (and therefore credible), and (iv) whether the messages generate few negative reactions. 

Messages that convey short-term risks appeared to be more persuasive to young adults in France than messages that convey long-term risks. Theoretically, this can be explained by temporal construal level and prospect theory, which states that people react to an event depending on losses or gains associated with their behavior and the temporal closeness of the outcomes [63,64]. Proximate adverse outcomes (short-term risks, e.g., alcohol coma and car accidents) appear more concrete and certain and therefore, are more persuasive, whereas more distant outcomes (long-term risks, e.g., risk of alcoholism and isolation) are perceived as more abstract, uncertain, and too far away. In this exploratory study, we found that prospect theory applied for most of the long-term warnings, except for the fear-inducing messages, typically on alcohol-induced brain damage or cancer, that motivate respondents to think about their drinking habits.

Warnings that displayed risks which young adults are exposed to (e.g., road accidents, sexual assault, and alcohol coma) appeared to be the most believable and impactful warnings in terms of raising concern and awareness. This is consistent with theoretical frameworks such as the theory of planned behavior, protection motivation theory, and the COM-B model [65,66,67]. These models have shown that cues for action (in our case, warning labels) can trigger behavioral change if they raise perceived susceptibility. The literature suggests that one way to increase perceived susceptibility (and thus the threat of personal risk) is to use a narrative message that contains information about a setting, characters, and their motivations [68] as this format makes it easier for people to imagine an event or construct a scenario for themselves, and thus increases their appraisal of risk likelihood. Narrative warnings are used in messaging on tobacco, where the testimonial format serves to increase awareness of risk susceptibility [69,70], but not on alcohol labels. It would be instructive for future research to explore how personal risk appraisals in young adults are affected by a warning displaying a testimonial from a young person who had developed an alcohol use disorder after engaging in binge-drinking behaviors for several years.

Our results also reveal that a lack of knowledge about certain alcohol risks can affect the perceived effectiveness and credibility of the warnings (e.g., on coma, cancer, and brain damage). One way to tackle this issue is to deploy mass-media campaigns alongside the launch of new alcohol warnings, as has been done for tobacco warnings; mass-media advertisements enhance the impact of tobacco warnings on knowledge about the health effects of smoking [71,72]. Furthermore, research has shown that warnings raising awareness around the fact that alcohol can cause cancer can increase support for alcohol policies [73], although this measure may be difficult to implement in France, where there have been only a rare, few campaigns on alcohol-use risks (other than drink–driving campaigns) targeted specifically at young people.

Negative reactions to warnings in general have been proven to undermine their impact [74]. Various criticisms of the warnings emerged in this study, including perceptions that the risk messages were ridiculous, exaggerated, irrelevant, odd, a threat to freedom (reactance), preachy, easy to avoid without having to change drinking habits (e.g., doing sport to offset the calorie intake from alcohol), and unpersuasive in terms of reducing alcohol consumption and avoiding binge drinking. The messages that generated most negative reactions were the ”innovative” messages, i.e., messages that focus on calorie intake, environmental issues, and manipulation by the alcohol industry. The negative reactions may be explained by context factors, i.e., drinking culture in France and the low acceptability of alcohol warnings among French people [19,75].

Concerning the warning format, most respondents welcomed the idea of adding pictograms to warnings: the pictogram–message format had a positive effect on the persuasiveness process (though attention, comprehension, risk perception, awareness, and susceptibility) while generating few negative reactions. Larger warnings combined with pictograms, thus, appears to be a promising route for developing messages that are easier to see, read, and understand; more impactful for a young adult audience; and ultimately, more likely to educate them and raise awareness of the risks related to alcohol use. The added value of pictograms could be explained by the elaboration likelihood model, which posits that people can be persuaded by the merits of the arguments in a message (central route of persuasion) or by factors around the central message that generate emotion (e.g., pictures, music, popular spokesperson, peripheral route of persuasion) [76]. Warnings in text-plus-pictogram format may have activated both routes (the peripheral route via the pictogram and the central route via the text).

Finally, it is important to note that some of the tested warnings generated negative reactions (avoidance) but none of them prompted the intended behavioral change (i.e., none of the participants reported subsequently wanting to reduce their drinking). Different reasons can explain this result. First, our research and other studies on alcohol warning labels suggest they are relevant as tools for raising awareness and knowledge about alcohol-related risks, but they need to be backed up by broader alcohol policy approaches and measures that change the positive norms around alcohol (such as a mass-media campaign and bans on alcohol advertising) and that prompt people to change their alcohol-use behaviors (such as bringing in price increases and measures restricting the availability of alcohol). The lack of effect on behavioral intentions observed may also be explained by the extended parallel process model [77], which states that when a threat and negative risks associated with a behavior are presented alone, without being combined with response efficacy and/or self-efficacy messages (e.g., positively worded messages phrased as a recommendation), people feel unable to escape the threat, and so the prevention message ultimately becomes ineffective in changing behaviors. Research has revealed that positive messages on cessation combined with negative warnings have an effective impact in the context of tobacco warnings [78,79]. It would be instructive for future research to test the effect of combining negative warnings on alcohol with positive messages on drinking less.

This lack of impact of warning on behavior, however, should be interpreted with caution. Longitudinal research on the effects of tobacco warnings has shown that avoidance behavior signals warning engagement [80] and is related to smoking cessation [81]. Longitudinal research could be conducted on alcohol warnings to test the relationship between warning engagement and drinking behavior.

This research has some limitations. First, it is essentially an exploratory study, which made it difficult, for example, to compare the reactions of the participants according to their profiles. Quantitative research could be conducted on a larger sample of young adults in France to gather a larger and more representative set of data and to analyze the impact of the warnings according to drinker profiles.

Another limitation is that the effects of the warnings were not measured on real behavioral responses but only on behavioral intentions. In addition, this study on alcohol-use warning content and formats provides data on young adults, whereas other alcohol-vulnerable populations (e.g., adults, women, pregnant women, etc.) also warrant similar investigation. Furthermore, the results of this study concern warnings inserted on advertisements. It would be interesting to investigate whether such warnings would be relevant for packaging. Our classification of warnings as health risks or social-cost risks may also be a limitation of this study, as some warnings have both a social-cost and a health-cost component (e.g., drunk driving accidents).

Despite these limitations, this study yielded valuable insight into the effect of alcohol warnings on young adults who live in a European country (here, France) where the social acceptability of drinking alcohol is high.

## 5. Conclusions

This research adds to the literature by exploring reactions to various warnings in young drinkers who live in a country where alcohol consumption rates are high, a context with limited previous research. As suggested by Davies et al. [82], it is important to address differences between countries to devise appropriately targeted alcohol harm reduction measures and to better inform what should be featured on public health labels. The results highlight that, despite living in a culture that is conducive to drinking alcohol, young adult drinkers in France are receptive to alcohol warnings that focus on specific and familiar risks. The results of this study provide public health actors with useful insights to help inform the current debate on European warnings and improve alcohol-use warning labels.

## Figures and Tables

**Figure 2 ijerph-20-06541-f002:**
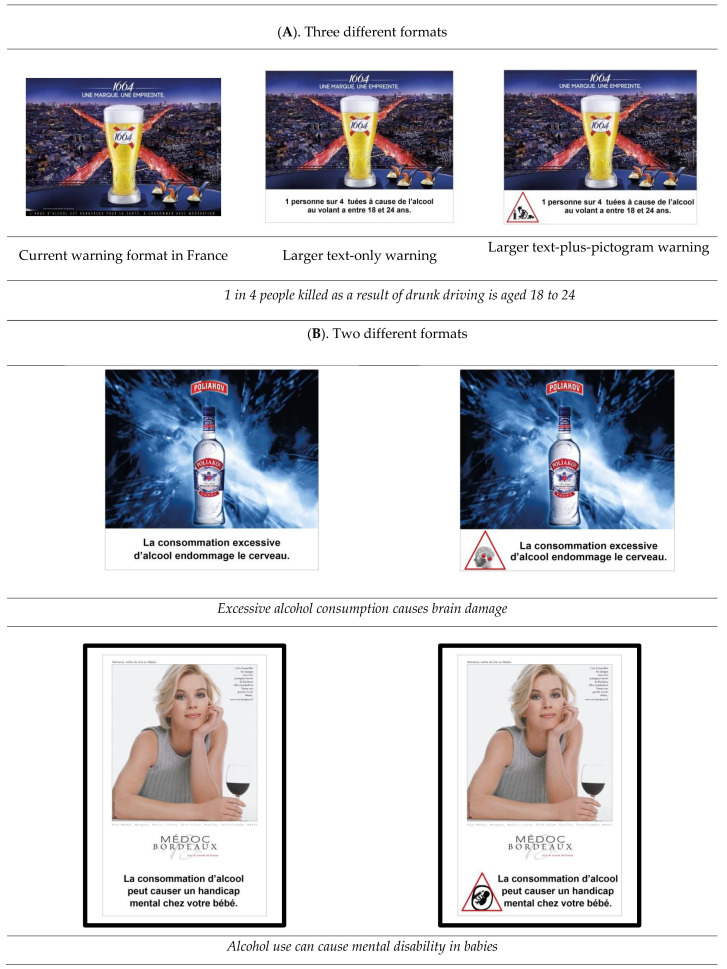
Examples of tested warning formats. **Legend:** Stimuli presented to the participants. Warnings displayed on real advertisements for beers, vodka, or wines. These images were created by an advertising agency solely for the purposes of this study. They are mock ads that use real brands. The brands did not participate or endorsed the study.

**Table 1 ijerph-20-06541-t001:** Description of the sample.

Participant	Gender	Age	Drinking Profile (AUDIT–C Score)	Highest Educational Attainment	Occupation
1.	M	24	High-risk drinker (7)	High-school qualifications	Student (social work)
2.	F	18	Moderate drinker (2)	High-school qualifications	Student (psychology)
3.	F	19	High-risk drinker (6)	High-school qualification	Student (social work)
4.	M	22	Moderate drinker (3)	University undergraduate	Student (finance)
5.	F	20	High-risk drinker (5)	University undergraduate	Student (business)/part-time helpline operator
6.	M	22	Moderate drinker (3)	University undergraduate	Student (education)/extracurricular activity leader
7.	M	19	High-risk drinker (6)	High-school qualifications	Student (information technology)
8.	F	22	Moderate drinker (1)	University undergraduate	Student (physics and chemistry)
9.	F	25	High-risk drinker (4)	High-school qualifications	Front desk operator (car rental agency)
10.	M	19	Moderate drinker (3)	High-school qualifications	Student (literature)
11.	M	20	High-risk drinker (5)	High-school qualifications	Student (business)
12.	F	19	Moderate drinker (2)	High-school qualifications	Student (business management)
13.	F	19	High-risk drinker (6)	High-school qualifications	Student (opticianry)
14.	F	25	Moderate drinker (2)	University undergraduate	Supermarket cashier
15.	M	20	High-risk drinker (5)	University undergraduate	Student (sports education)/part-time lifeguard
16.	F	21	Moderate drinker (2)	University undergraduate	Student (customer relations/sales)
17.	M	24	High-risk drinker (7)		Toolmaker
18.	F	23	High-risk drinker (5)	University undergraduate	Sales rep (civil engineering)
19.	F	19	Moderate drinker (2)	High-school qualifications	Student (management support)
20.	M	20	High-risk drinker (7)	High-school qualifications	Student (mechanical design and industrialization)
21.	M	25	Moderate drinker (3)	Master’s degree	Student (communications)
22.	F	22	High-risk drinker (5)	University undergraduate	Social worker
23.	F	22	Moderate drinker (2)	University undergraduate	Specialist educator
24.	M	24	Moderate drinker (1)	High-school qualifications	Mail carrier in a purchasing department
25.	M	25	Moderate drinker (3)	High-school qualifications	Unskilled worker

## Data Availability

Data summaries can be provided upon reasonable request from the authors.

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
