# Peer review of "How Do Young Adult Drinkers React to Varied Alcohol Warning Formats and Contents? An Exploratory Study in France"

_ijerph, 2023, doi:10.3390/ijerph20156541_

Round 1

Reviewer 1 Report

Dear authors

This is a fairly interesting paper that can be published after a few alterations.

It is a very interesting subject, as alcohol use in ypung adults is still little studied investigated. Any advance in that area is, thus, welcome.

I present some suggestions that I would like you to consider, in order to the article to be published:

Line 40: wthat did you mean with "fifth third"? I think it would be easier to understand indicating this ratio with the corresponding percentage.

Line 40: per capita. I think you should replace in the whole manuscript these two words with te latin expression "pro capite" 

Line 61: "young French adult". I suggest to modify the sentence as "French young adults"

Moreover, I would like to suggest an editing of English language and a more concise presentation of the results.

I would like to suggest an editing of English language.

Reviewer 2 Report

Thank you very much for the opportunity to review the manuscript entitled “How do young adult drinkers react to varied alcohol warning formats and contents? An exploratory study in France, a country with a strong drinking culture”. I think that the authors show an interesting article, since it addresses the responses of young drinkers to different warning stimuli, which makes it closely related to the editorial line of the magazine. In addition, a qualitative study is carried out, which provides added value.

I believe that the manuscript could be accepted with some minor changes, so the following recommendations are proposed:

- At the end of the introduction (which is appropriate), it is recommended that the authors formulate a research problem and define the associated hypotheses. Subsequently, they must respond to these hypotheses (acceptance or not) in the conclusions of the study.

- Material and method: select method for the selection of participants (sample). Were there any inclusion or exclusion criteria? Detail, as it will improve the study methodology.

- The authors state that they used the AUDIT test to identify the consumption patterns of the respondents. On the one hand, it is recommended that they provide the scores obtained for each of the subjects (and not just their profile). On the other hand, it would be advisable for the description of the test to be more specifically detailed in a section called "Instruments".

- Add the code of the ethics committee that approved the article.

- The results are provided in a fairly descriptive way (it is correct, but it can be improved). It would be interesting if the authors carried out a meta-category analysis of the interviews. Furthermore, since they use NVIVO, they could generate some type of figure that illustrates the most relevant data, such as a hierarchical map, word cloud or diagram (at the interest of the authors). This will make the interpretation of the results much more visual and will make it easier for the reader to understand.

- The authors point out the limitations of their study appropriately, which is appreciated.

- Include the response to hypotheses previously developed in the conclusions section.

Reviewer 3 Report

Thank you for the opportunity to review this paper. It investigates a timely topic and will add new insights to the literature on alcohol health warnings in a non-English speaking country, focusing on evaluation on variety of different possible messages. Overall, the paper is already very comprehensive and well developed. However, before acceptance, I would like the authors to take in consideration the following comments: 

Title:

1)     the title refers to France with a “strong drinking culture”. This expression is somewhat vague – what does strong drinking culture supposed to mean? Is there any expression that could be a bit more precise? The title also does not mention the research refers to warnings on advertisement. I think that should be mentioned either in title or the abstract.

Abstract:

1)     The claim “and very few studies have been led in countries with very high  per-capita alcohol consumption” can be disputed on the grounds that the Anglosphere countries that are previously referred to have relatively high per capita alcohol consumption. Is it possible to find some other characterization?

2)     Likewise, the abstract refers to “pro-drinking culture environment”. Is it possible to find an expression that is more nuanced? Maybe something along the lines “high social acceptability of alcohol/drinking”?

3)     Perhaps mention the year of data collection (2017) already in the abstract, for the reader to be aware these interviews were conducted before the current policy discussions that are happening on EU and national levels.

4)     In text: “ Larger text-and-pictogram warnings were the most effective to catch attention and increase awareness” – replace with “were considered” or similar phrasing, as you did not measure effectiveness with qualitative study

5)     Also here perhaps mention that those were warnings on advertisements – or is that intentionally left out?

Introduction

1)     In sentence: “France, where this re search takes place, has the fifth third largest alcohol consumption per capita in Europe (11.4 litres per year per capita, 15 and over; [3]) and registers 41,000 alcohol-related deaths per year [4].” - refer to the year of the data collected if possible.

2)     Youth drinking is a major public health issue: about 50% of 18– 42 34-year-olds report at least one heavy drinking episode per month [5] – is that figure for France or in general?

3)     “In the introduction, you refer to France as country with “big drinking culture” couple of times   can you find a more suitable expression than “big drinking culture”? In the first instance you use it, you could also rephrase in this direction– “due to lack of suitable alcohol control policies and strong influence of the economic operators, the consumption and social acceptability of alcohol is high”. Speaking of drinking culture too much might also imply the role of tradition, as opposed to the role of economic operators and commercial determinants of health. Perhaps look at this  paper and if it contains anything relevant for you in characterization of the drinking culture in France? Gordon, R., Heim, D., & MacAskill, S. (2012). Rethinking drinking cultures: A review of drinking cultures and a reconstructed dimensional approach. Public health, 126(1), 3-11.

4)     In line 66, there is a redundant “)”

5) In the sentence “ However, these warnings have proven ineffective at grabbing attention and informing French young adults, as they are not prominent enough, too small, too old, and the text is unclear” - are you referring to mandatory warnings on advertisements or warnings on labels (or both) ?

General comment: a new study was just published that also examines perceptions of different labels by young people; is that useful for you to add to your literature review (which is already quite comprehensive) and interpret your results? López-Olmedo, N., Muciño-Sandoval, K., Canto-Osorio, F. et al. Warning labels on alcoholic beverage containers: a pilot randomized experiment among young adults in Mexico. BMC Public Health 23, 1156 (2023). https://doi.org/10.1186/s12889-023-16069-w

Methods

1)     line 121: I would briefly add already here how were moderate and high risk drinkers defined based on the AUDIT score, and that participants with highest consumption were screened out (even though you mention this information in the table later on, but it took me a while to get to the bottom of the table, and not all the readers will read the table).  

2)     Line 139 (Tested warnings): In this paragraph, please note early enough that those warnings are presented in Figure on the next page. I started reading and was wondering how the messages look like while I was reading the rationale, and only noticed the figure later.

3)     Do you have any indication of the education level / other indicator of SES of the participants? Can you include any information on that?

4)     Line 164: “The 12 warnings were presented to respondents in different formats: first, the current (small) French format vs the larger text-only warning format vs the larger text-plus-pictogram (Figure 2.A), and second, the larger text-only warning format vs the larger text-plus-pictogram (Figure 2.B). All warnings were presented to respondents on printed boards. “  - Can you rewrite this sentence to be clearer? It’s clear once you look at the figures, but not before. Also, what you provide a brief rationale for the sequence of shown warnings (also as described in procedure)?

Results

1)     The results section is quite long, with a lot of quotes, but not very concrete summary or conclusions, it feels more like a listing of different opinions. Maybe it would be helpful if a table with key results/themes would be added, perhaps by warning topic? This way reader can get a succinct impression of the key outcomes.

2)     In section 3.2: “Compared to social-cost warnings, health warnings lead to greater knowledge and awareness among the participants:”  - you cannot speak about impact as it’s a qualitative study. Can you rephrase this to better reflect lack of causality?

Discussion

1)     The discussion is very comprehensive. One thing that could still be considered is reflection whether the findings regarding warnings on advertisements can be applied also to warnings on labels.

Reviewer 4 Report

The authors have addressed well the research question, which would be contributive to the literature.
